# The Effects of Pilates Exercise Training Combined with Walking on Cardiorespiratory Fitness, Functional Capacity, and Disease Activity in Patients with Non-Radiologically Confirmed Axial Spondylitis

**DOI:** 10.3390/jfmk8040140

**Published:** 2023-10-04

**Authors:** Eleni Zaggelidou, Athina Theodoridou, Vassiliki Michou, Helen Gika, George Panayiotou, Theodoros Dimitroulas, Evangelia Kouidi

**Affiliations:** 1Sports Medicine Laboratory, School of Physical Education & Sport Science, Aristotle University, 57001 Thessaloniki, Greece; elzaggel@gmail.com (E.Z.); vasilikimichou@yahoo.gr (V.M.); 24th Department of Internal Medicine, Hippokrateion General Hospital of Thessaloniki, School of Medicine, Aristotle University, 54642 Thessaloniki, Greece; atheodori@yahoo.gr (A.T.); dimitroul@hotmail.com (T.D.); 3Forensic Medicine and Toxicology, Aristotle University of Thessaloniki Medical School, 54124 Thessaloniki, Greece; gkikae@auth.gr; 4S Laboratory of Exercise, Health and Human Performance, Applied Sport Science Postgraduate Program, Department of Life Sciences, School of Sciences, European University Cyprus, 2404 Nicosia, Cyprus; g.panayiotou@euc.ac.cy

**Keywords:** non-radiological axial spondylarthritis, Pilates exercise, functional capacity, disease activity

## Abstract

The objective of the study was to examine the effects of Pilates exercise training combined with walking on cardiorespiratory fitness, functional capacity, and disease activity in patients with non-radiologically confirmed axial spondylitis (nr-axSpA). Thirty patients with nr-axSpA (seven women (90%), with a mean age of 46.07 ± 10.48 years old and C-reactive protein (CRP) 2.26 ± 2.14 mg/L) were randomly divided into two groups: A (*n*_1_ = 15 patients) and B (*n*_2_ = 15 patients). Group A followed a 6-month home-based Pilates exercise training program, while Group B remained untrained until the end of the study. A cardiopulmonary exercise test (CPET), timed up and go test (TUG), five times sit-to-stand test (5×STS), sit-and-reach test (SR), back scratch test for the right (BSR) and the left arm (BSL), Bath Ankylosing Spondylitis Disease Activity Index (BASDAI), and Ankylosing Spondylitis Disease Activity Score (ASDAS) were applied to all patients, both at the beginning and at the end of the study. After 6 months, Group A showed higher values in exercise time by 37.41% (*p* = 0.001), higher peak oxygen uptake (VO_2_peak) by 25.41% (*p* = 0.01), a higher ratio between oxygen uptake and maximum heart rate (VO_2_/HRmax) by 14.83% (*p* = 0.04), and higher SR by 18.70% (*p* = 0.007), while lower values were observed in TUG by 24.32% (*p* = 0.001), 5×STS by 12.13% (*p* = 0.001), BASDAI score by 20.00% (*p* = 0.04) and ASDAS score by 23.41% (*p* = 0.03), compared to Group B. Furthermore, linear regression analysis showed a positive correlation in Group A between BASDAI and 5×STS (r = 0.584, *p* = 0.02), BASDAI and TUG (r = 0.538, *p* = 0.03), and ASDAS and 5×STS (r = 0.538, *p* = 0.03), while a negative correlation was found between BASDAI and VO_2_peak (r = −0.782, *p* < 0.001), ASDAS and SR (r = −0.548, *p* = 0.03), and ASDAS and VO_2_peak (r = −0.659, *p* = 0.008). To sum up, cardiorespiratory fitness, functional capacity, and disease activity improved after a long-term Pilates exercise training program in patients with nr-axSpA.

## 1. Introduction

Axial spondylarthritis (axSpA) is a continuous inflammatory disease that primarily affects the spine and the sacroiliac joint, which connects the lower spine to the pelvis. There are two types of axial spondylarthritis: non-radiological axial spondylarthritis (nr-axSpA), in which symptoms of axSpA are present but joint damage is not visible on the X-ray, and ankylosing spondylitis (AS), in which signs of axSpA are present and joint damage is visible on the X-ray [1]. Moreover, it has been found that there are similarities between the two types of ankylosing spondylitis with similar human leukocyte antigen B27 (HLA-B27) subtypes [2]. In contrast, according to a recent review, unlike axSpA, which is male-predominant [3], nr-axSpA is equally or slightly more prevalent in women than in men [4].

Patients with axSpA have, on average, lower weekly levels of exercise, while their aerobic capacity is significantly lower compared to healthy individuals [5,6]. For patients with extensive radiographic involvement, physical therapy and exercise are crucial components of managing their condition, as medication alone has never been considered sufficient [7,8]. In patients with axSpA, it has been found that a properly structured exercise program significantly improves the spine’s mobility, reduces disease activity, and enhances functional capacity [7,9]. In addition, despite the favorable benefits of exercise training in patients with ax-SpA, signs such as joint pain and fatigue may reduce their level of physical activity and participation in long-term exercise training [10].

Even though different types of exercise have been previously applied in the axSpA population (such as aerobic exercise, respiratory exercises, Pilates, stretching, and flexibility exercises [11,12]), Pilates training tends to be more and more practiced in patients with AS and axSpA [12,13,14]. It is already known that Pilates training effectively increases physical capacity, decreases pain, and improves spinal mobility and respiratory function in patients with AS [15,16]. In addition, Pilates training combined with aerobic exercise leads to better results in disease activity, functional capacity, and muscle strength than aerobic exercise alone [17]. Nevertheless, there are limited studies concerning the effects of exercise programs on patients with nr-axSpA. Thus, we hypothesized that long-term home-based Pilates exercise training for patients with nr-axSpA may lead to favorable improvements in cardiorespiratory fitness, functional capacity, and disease activity in patients with nr-axSpA.

## 2. Materials and Methods

### 2.1. Participants

Patients with nr-axSpA without systemic medication were recruited and screened for eligibility from Rheumatology clinics and from the Hippokrateion General Hospital of Thessaloniki, Greece. Inclusion criteria included age (adults less than 65 years old), diagnosis with nr-axSpA from a rheumatologist in an academic hospital, fulfilling the ASAS 2009 criteria for nr-axSpA [18], having only symptoms from the axial skeleton (spine, shoulders, and hips) and no peripheral arthritis and not receiving any medication. Patients with other reasons for back pain, peripheral arthritis, extraarticular manifestations of SpA, chronic comorbidities, such as diabetes mellitus, chronic lung or kidney disease, cardiac arrhythmias, unstable hypertension, and severe anemia (Hemoglobin (Hb) <12g/dL for females and <13g/dL for males), and those receiving medication that included biological agents, mainly anti-Tumor Necrosis Factor (anti-TNF), Disease-Modifying Antirheumatic Drugs (DMARDs) such as Sulfasalazine, painkillers, or anti-inflammatory drugs were excluded from the study. Patients who met the inclusion criteria and willingly chose to participate were enrolled in our study.

### 2.2. Study Design

Before enrollment, all participants were given complete study information and required to provide written informed consent to participate. Clinical examination included anthropometric measurements (such as body weight, height, and calculation of BMI according to formula kg/m^2^ [19]), review of their medical history (clinical profile and need to use painkillers/anti-inflammatory drugs), and blood sampling. Then, patients with nr-axSpA who met the above inclusion criteria underwent a clinical and functional capacity examination at the Sports Medicine Laboratory of the Department of Physical Education and Sports Science of the Aristotle University of Thessaloniki, Greece. The disease’s activity and treatment effectiveness were assessed with Bath Ankylosing Spondylitis Disease Activity Index (BASDAI) and Ankylosing Spondylitis Disease Activity Score (ASDAS). Furthermore, all participants underwent cardiopulmonary exercise testing (CPET) to estimate peak oxygen uptake (VO_2_peak), anaerobic threshold, and pulmonary ventilation (VEmax) and functional capacity tests, such as timed up and go test (TUG), five times sit-to-stand test (5×STS), and sit-and-reach test (SR) to assess overall functionality, while back scratch test for the right (BSR) and the left arm (BSL) was applied to evaluate patients’ flexibility. After the initial measurements, patients were randomly assigned to either Group A (exercise group) or Group B (control group). For the randomization method, the https://www.randomizer.org website was used (assessed on 29 November 2022). Participants and research assistants were blinded to the patients’ group allocations. At the end of the 6-month trial, clinical and functional capacity measurements were repeated. The same researcher who was unbiased toward group assignment carried out all tests. This randomized clinical trial was anonymous, and the General Data Protection Regulation (GDPR, Regulation E.E. 2016/679) was applied. The study’s protocol was approved by the Ethics Committee of School of Physical Education & Sport Science, Aristotle University, Thessaloniki, Greece (Approval number EC-45/2021). During the study, none of the patients were receiving any medication. The clinical trial began in September 2022 and ended in April 2023.

### 2.3. Sample Size Estimation

To determine the necessary sample size, we referred to a previous study by Yentür et al. [20] that focused on patients with rheumatoid arthritis. We hypothesized significant differences between Groups A and B in disease activity total score and functional capacity indices. After conducting a two-tailed test of significance with a significance level of *p* < 0.05, we determined that to achieve a power of 80%, we would need 15 individuals per group (assuming a 20% dropout rate). Thus, a total of 30 participants were randomly assigned (post hoc power analysis ranged between 68.0 and 71.5%) into two equal groups (Group A = 15 participants and Group B = 15 participants). The sample size of 30 participants was based on the nr-axSpA population from the Rheumatology clinics and the Hippokrateion General Hospital of Thessaloniki.

### 2.4. Blood Analysis

At baseline and at the end of the 6-month study, blood samples were taken from participants who were fasted for at least 12 h. The levels of the Erythrocyte Sedimentation Rate (ESR) and C-reactive protein (CRP) were evaluated to detect the levels of inflammation. The Westergren method was used for the ESR analysis. Normal values for ESR are as follows: ≤15 mm/h for males less than 50 years old, ≤20 mm/h for males over 50 years old, ≤ 20 mm/h for females less than 50 years old, and ≤30 mm/h for females over 50 years old [21].

CRP levels were quantified through the immunoassay method. CRP measurement is commonly used to differentiate many acute or chronic conditions in detecting the presence or worsening of inflammatory processes and monitoring the response of inflammatory diseases to remedy. It is a more reliable indicator than ESR for evaluating inflammatory conditions. CRP values are interpreted as follows [22]:Less than 0.3 mg/dL: Normal.Between 0.3 and 1.0 mg/dL: Normal or minor elevation.Between 1.0 and 10.0 mg/dL: Moderate elevation.More than 10.0 mg/dL: Notable elevation.More than 50.0 mg/dL: Severe elevation.

### 2.5. Cardiorespiratory Exercise Testing

All patients underwent a maximum exercise test on a treadmill using the Bruce protocol, with simultaneous ergospirometry via an Ultima CPX Series ergospirometer (Medical Graphics Corporation, Saint Paul, MN, USA) to estimate time to fatigue, metabolic equivalents (METs), peak oxygen uptake (VO_2_peak), respiratory exchange ratio (RER), oxygen pulse (ratio between oxygen uptake (VO_2_) and maximum heart rate (HRmax) (VO_2_/HRmax)), ventilatory equivalents for oxygen (VE/VO_2_max), ventilatory equivalents for carbon dioxide (VE/VCO_2_max), heart rate, and systolic blood pressure (SBP) and diastolic blood pressure (DBP) at rest and exercise. The endpoint of the cardiorespiratory test was considered as either a RER ≥1.10 [23] or the oxygen plateau (change in oxygen consumption less than 150 mL/min during the last minute of exercise) observed during maximal exercise [24,25].

### 2.6. Timed up and Go Test

TUG trial is a well established and reliable test to evaluate a participant’s overall mobility and functionality. According to Kear et al. [26], TUG times significantly differed between the ages of 20 and 59 (F = 6.579, *p* = 0.001). Slower times were recorded in the 50 s compared to the 20 s (*p* = 0.001), 30 s (*p* = 0.001), and 40 s (*p* = 0.020) [26]. Participants were instructed to sit on a 46 cm highchair, with their knees at a right angle (90 degrees) and hands on the chest, and at the “go command” of the examiner, to stand up, walk a 3 m distance, and return to the initial position. The examiner measured the test execution time, which was recorded in seconds. A faster time suggests better functional performance, and a score of ≥13.5 s is used as a cut-off point to determine those at increased risk of falls in the community setting [27].

### 2.7. Five Times Sit-to-Stand Test

The 5×STS is a low-cost and valuable tool for assessing sitting and standing ability. Bermejo et al. [28] showed that the 5×STS has excellent reliability in adults (ICC = 0.937, *p* < 0.001). Based on the study of Bohannon et al. [29], normal references for 5×STS range from 6.0 s (20–29 years) to 10.8 s (80–85 years).

Participants were asked to sit in an armless chair of 45 cm. The instructions given to every participant were to cross their arms over their chest and sit with their back against the chair’s upright backrest. On the “go command”, each participant had to stand up with an upright trunk with hips and knees extended and then sit down five times in a row. Data were collected in seconds. The less time a participant performed, the better the score was.

### 2.8. Sit-and-Reach Test

The sit-and-reach (SR) is widely used to measure hamstring flexibility [30,31]. Hamstring flexibility was assessed from a standing position with the legs extended, the heels attached to the specially designed box (length 53.0 cm and height 32.5 cm), and the palms of the hands placed on top of each other. The participant had to bend the trunk forward and reach with the palms as far as possible, staying in this position for two seconds without bending the knees. After a total of three attempts, the best performance was recorded. Performance was noted in centimeters as the distance from the middle finger to the measuring scale on the box. Before the start of the measurement, the researcher gave the appropriate execution instructions, and a trial attempt was carried out.

### 2.9. Back Scratch Test

The back scratch (BS) test was used to assess the flexibility of the shoulder joint and shoulder arch on both the right (BSR) and left sides (BSL) [32]. According to the BS protocol, participants were instructed to start the trial upright, place the left arm on the arm/hand on their lower back, and then move it up their spine toward their head. The right arm/hand should be placed behind the neck and moved down in the back. The aim of the test was to put the long finger of each hand as close to the other or to overlap the other hand as much as possible. If the fingertips overlapped, the score was noted as a positive symbol (+X cm); if the fingertips of both the right and left hand touched each other, then a score of 0 cm was recorded, and if they did not reach it, the score was negative (−X cm) [32,33].

### 2.10. Bath Ankylosing Spondylitis Disease Activity Index

The BASDAI questionnaire is a low-cost, self-reported, easy way to evaluate disease activity. Recording disease activity is vital to quantify inflammation and optimize prognosis in patients with Ankylosing Spondylitis Disease (ASD) [34]. BASDAI consists of 6 questions regarding fatigue, back pain, peripheral pain/swelling, enthesitis, severity, and duration of morning stiffness. Each question is rated from 0 to 10. For the calculation process, the researcher first calculated the average score between the two last questions regarding morning stiffness and then added the score to the mean score of the remaining four elements. The average score of the total items was between 0 and 10 [15,34]. A mean score >/ = 4 implied high disease activity and a criterion for starting biological therapy in patients with ASD. In contrast, a mean score <4 implied lower disease activity [35].

### 2.11. Ankylosing Spondylitis Disease Activity Score

The ASDAS questionnaire is also a self-reported and low-cost tool to evaluate disease activity in patients with ASD. According to the literature, there are two formulas for ASDAS calculation: the ASDAS-CRP and ASDAS-ESR. The first is considered the preferred version, and the second is the alternative version [18,36]. In our study, we used the ASDAS-CRP formula that was calculated as follows: 0.12 × back pain + 0.06 × duration of morning stiffness + 0.11 × patient global + 0.07 × peripheral pain/swelling + 0.58 × Ln(CRP+1) [15]. CRP was measured in mg/L. The range of the four remaining items (back pain, duration of morning stiffness, patient global and peripheral pain, and swelling) was between 0 and 10. In addition, it is notable that there are three cut-offs for evaluating the ASDAS score: a score of <1.3 implies between inactive disease and moderate disease activity, <2.1 implies between moderate and high disease activity, and >3.5 indicates between high disease and very high disease activity [36].

### 2.12. Pilates Exercise Training

Patients of Group A, due to lack of free time, inability to move in the sports med laboratory for the exercise sessions, and to preserve their autonomy, followed a 6-month Pilates-based exercise training program at home through the Zoom platform (San Jose, CA, USA): “https://zoom.us (accessed on 1 October 2022)”. The Pilates method is a low-intensity exercise that cultivates body awareness, strengthening the abdominal and back muscles and is, therefore, used to improve stability and support in the body. After the initial evaluation, the exercises were adjusted according to the needs and specifications of each person, creating a personalized exercise program. The program included breathing and correct body position exercises, activities in a stable position with an emphasis on the abdominals, exercises from a prone position, body stretches, and proprioceptive and trunk stabilization exercises. The equipment used were resistance bands, the Pilates ring (magic circle), and weights of 1 kg. Patients exercised using a 1:1 (trainer/patient) supervision ratio three times weekly on non-consecutive days. Thus, the personal trainer had the ability to demonstrate the exercises, to explain proper techniques, and to ensure that each patient executed the exercises correctly. Each session lasted 60 min with a moderate intensity, which was kept according to the subjective scale of Borg, between 13 and 14. In detail, the program was divided into three parts (Figure 1): A warm-up part (10 min), the Pilates exercises (40 min), and a cool-down part (10 min). All participants followed the same exercise protocol.

Participants were asked to maintain their video and sound during the exercise session to ensure that each patient performed the exercises correctly. Moreover, voice feedback was given during each session to enhance each patient’s exercise performance. During the first two months, the repetitions of each exercise ranged from 4 to 6 and gradually increased from 6 to 10. Between the exercises’ sets, there was a rest period from 10 s to 15 s, which varied as the patient gained more adaptations and awareness of the exercise, gradually reducing the suspension of the exercises. The goal was the sequential flow from one exercise to another according to Pilates’ principles. During the exercise period, the number of all repetitions in the sets was increased by 10 in each exercise. There was also an increase in the number of activities in each session depending on the lesson’s goal and the patient’s level. The exercises that were used were as follows:One Leg Stretch: Patients were asked to lie in a supine position on the mat and to pull one knee into their chest, then inhale and start to bend the out-stretched leg and straighten the bent leg. At the end of this exercise, patients exhale.Double Leg Stretch: Firstly, patients in the same supine position on the mat were asked to bend their legs, with their feet off the floor, and grab both knees. Secondly, they had to lift both shoulders off the floor, extend their arms toward their ears, and simultaneously raise both legs to a 45-degree angle from the floor. Thirdly, they had to bend their knees and bring their chin toward their chest while hugging them. Lastly, patients had to extend their upper and lower limbs and repeat until the set was completed.Shoulder Bridge: Firstly, patients were instructed to lay on their backs with their knees bent, heels lined with their bottom, and arms rested by their sides. Secondly, they were instructed to take a deep breath in; as they exhaled, they flattened their lower back to the floor as though they were lifting their tailbone to the ceiling. They had to visualize each vertebra leaving the floor one by one until they were weight-bearing through their shoulders.Chest Lift: In this exercise, patients were asked to lie down, keep their knees bent, their back and feet flat, and their hands supporting their heads. Then, they had to lift their shoulders and squeeze their abdominal muscles. They were instructed to hold this position for 1 to 2 s and then relax by returning to the initial position.Hundreds: In this exercise, patients were asked to lie on their backs with their knees bent and legs parallel to the floor, lift their shoulders off the mat, and extend their upper and lower limbs. Then, they were instructed to inhale for 5 s and exhale for 5 s while pumping arms. To achieve 100, they had to act 10 times.One Leg Circles: Patients were instructed to lie on their back, with their arms down and by their side, pelvis in a neutral position, and core engaged, to extend the right leg toward the ceiling and the left leg along the mat. They had to inhale to prepare, exhale, and circle the right leg away from the midline, keeping the leg extended.Spine Stretch: In this exercise, patients were instructed to sit up tall as if their back was against a wall. Legs should be out in front of them and opened about shoulder distance apart. Knees and toes will be pointed to the ceiling, and heels will reach away from them to create length and oppositional energy. They had to lift their arms in front of them, with fingertips reaching, palms down. Then, they had to roll their shoulder blades down their back to create space between the neck and ears. After a deep inhale, they were asked to exhale as they curled their head, neck, and upper spine down the imaginary wall while pulling their abs in. Moreover, they had to stretch forward as if bending over a round barrel toward their toes. Next, they had to inhale as they began rolling back up the “wall”, starting with their tailbone, lower back, upper back, neck, and head, returning to the starting position feeling taller than before.Spine Twist: Initially, patients were asked to sit up tall on their feet and pull their abdominals in to support their upper body. Then, they had to flex their feet, reach their heels, and extend their arms directly out to the sides, keeping them even with their shoulders so there was one long line from fingertip to fingertip. In addition, patients were asked to exhale as they imagined a line running straight up through the middle of their body, turning their torso and head on that central axis and getting taller as they twisted. The movement is a two-part pulse where they had to exhale to twist halfway and then exhale again to turn as far as possible.Hip Twist: Firstly, patients were asked to lie on their backs and bend their legs, keeping their knees and feet parallel at hip-width apart and their arms by their sides. Secondly, they had to exhale, rotate their hips, and slowly open one leg outwards with their knee reaching the mat. Thirdly, they had to inhale and bring their knee back in line with their hip (1 repetition with the same leg). Lastly, they were instructed to keep their knee at a consistent angle and in line with their opposite knee as they rotate in their hip joint. Also, they had to maintain a stable pelvis and use their adductors to return the knee to the starting position.Swimming: Patients were asked to lie prone with extended upper and lower limbs. Then, they had to raise both arms and legs off the mat and lift their head and chest. Lastly, they were instructed to flutter their arms and legs and keep alternating sides for the entire set.Standing Side Bend: Initially, patients were asked to stand with their feet shoulder-width apart and put their left hand behind their head and their right hand at their side. Secondly, they had to bend to the right and lower their right hand toward the floor. Lastly, they had to return, switch sides, and repeat.Cat Stretching: Firstly, patients were asked to start with four-point kneeling, with hands underneath their shoulders and knees underneath their hips. Secondly, they were instructed to gently flex their neck by dropping their chin toward their chest while arching the rest of their spine into a curve. Thirdly, they had to slowly move into the opposite position by lifting their head upwards and extending their neck while allowing the rest of their spine to drop down into an extended position.

Finally, in conjunction with the program, participants were instructed to walk for about 30 min and up, with moderate intensity, i.e., at 50–70% of the predicted VO_2_ peak achieved during the Bruce cardiorespiratory test, on the remaining days when they did not have a zoom session. Walking sessions were monitored via a personnel diary, in which every patient described the time and frequency of walking. Group A adhered closely to the exercise program and participated in 85% of the scheduled sessions.

### 2.13. Statistical Analysis

The IBM SPSS 27 statistical package was used for analysis and results extraction (IBM Corp. Released 2020. IBM SPSS Statistics for Windows, Version 27.0. Armonk, NY, USA). The normal distribution of the variables was tested with the Kolmogorov–Smirnov test. Results are expressed as means ± standard deviation for normally distributed variables. The degree of variability among observers—both within the same observer and between different observers—was measured using the intraclass correlation coefficient (CCI) along with 95% confidence intervals (CI). Two-way ANOVA tested mean differences over time and between the two groups with repeated measures. *t*-test for independent samples was also applied to examine differences between groups. Linear regression was used to study the association between variables. The level of statistical significance was set at *p* < 0.05.

## 3. Results

### 3.1. Study Population

Of the 43 patients invited to participate in the study, 30 gave their written consent and underwent baseline measurements. Thus, after initial screening, a total of 30 patients were randomly assigned to either Group A (exercise group) or Group B (control group). The flow chart of the study is represented in Figure 2, and the patients’ baseline characteristics are shown in Table 1.

### 3.2. Cardiopulmonary Exercise Testing

Initially, there were no statistically significant differences between the two groups. At the end of the study, Group A showed favorable improvements in exercise time (increase by 26.46%, *p* = 0.001), METs (increase by 22.70%, *p* = 0.02), VO_2_peak (increase by 19.48%, *p* = 0.04), VO_2_/HRmax ratio (increase by 9.85%, *p* = 0.04), resting HR (decrease by 4.37%, *p* = 0.02), resting DBP (decrease by 4.76%, *p* = 0.03), exercise HR (increase by 7.85, *p* = 0.03), and exercise SBP (decrease by 5.15%, *p* = 0.04). Moreover, inter-group changes at the end of the study revealed that Group A statistically increased exercise time by 37.41% (*p* = 0.001), METs by 28.65% (*p* = 0.008), VO_2_peak by 25.43% (*p* = 0.01), and VO_2_/HRmax ratio by 14.83% (*p* = 0.04), while decreased resting HR by 5.50% (*p* = 0.001) compared to Group B (Table 2).

### 3.3. Functional Capacity and Disease Activity Results

At baseline, there was no statistically significant difference in functional capacity and spondylitis disease activity indicators between the two groups. After the 6-month Pilates program, Group A showed a significant decrease in the TUG by 16.47% (*p* = 0.001), 5×STS performance by 16.15% (*p* < 0.001), BASDAI score by 22.99% (*p* = 0.04), and ASDAS score by 27.44% (*p* = 0.04), while an increase in the SR by 15.46% (*p* = 0.001), BSR by 62.17% (*p* = 0.03), and BSL by 29.37% (*p* = 0.04) was noticed. In contrast, Group B did not demonstrate favorable changes during the follow-up period. Moreover, inter-group changes at the end of the study showed that Group A had statistically decreased scores in BASDAI by 20.00% (*p* = 0.04) and ASDAS by 23.41% (*p* = 0.03), decreased time in the TUG by 24.32% (*p* = 0.001) and 5×STS by 12.13% (*p* = 0.001), and increased performance in the SR by 18.70% (*p* = 0.007) compared to Group B (Table 3).

### 3.4. Linear Regression Analysis

At the end of the study a positive correlation was noticed in Group A between BASDAI and 5×STS (r = 0.584, *p* = 0.02), BASDAI and TUG (r = 0.538, *p* = 0.03), and ASDAS and 5×STS (r = 0.538, *p* = 0.03), while a negative correlation was found between BASDAI and VO_2_peak (r = −0.782, *p* < 0.001), ASDAS and SR (r = −0.548, *p* = 0.03), and ASDAS and VO_2_peak (r = −0.659, *p* = 0.008) (Figure 3).

## 4. Discussion

The aim of the study was to examine the effects of Pilates exercise training on cardiorespiratory fitness, functional capacity, and disease activity in patients with nr-axSpA. The main result of this study is that a long-term personalized home-based Pilates exercise program combined with walking improves cardiorespiratory fitness and functional capacity. At the same time, disease activity statistically decreased in patients with nr-axSpA.

Evidence of the efficacy of physiotherapy in the “early” stage of AS and in nr-axSpA is recent since the implementation of the Assessment of Spondylarthritis International Society (ASAS) criteria in 2009 [37]. According to ASAS, physical exercise is still regarded as a non-pharmacological treatment not only for AS [15] but for other rheumatic and musculoskeletal diseases [38,39]. According to the ACR-SPARTAN-SAA AxSpA Guidelines, supervised exercise interventions are highly recommended compared to passive physical therapy interventions, such as massage or heat, for patients with AS and nr-axSpA [40,41]. However, a combination of biological therapy and physical activity has synergistic effects, while cardiovascular risk and pain are highly eliminated, and functional and respiratory capacity are improved in patients with axSpA [42,43]. Systematic exercise, such as aerobic or Pilates exercise, has been proven to have anti-inflammatory and pain relief effects. The main anti-inflammatory mechanisms are lower levels of circulating adipokines due to visceral fat reduction, removal of cytokines (i.e., interleukin 6 (IL-6)) through skeletal muscle fiber contraction [44], higher circulating levels of adiponectin, and decreased expression of Toll-like receptors (TLRs) on monocytes and macrophages [45].

Even though the effects of exercise in axSpA have been previously demonstrated, to our knowledge, the first study that evaluated the effects of long-term home-based exercise training in both patients with AS and nr-axSpA was published in 2016 [46]. The results of this study showed that patients with nr-axSpA did not have statistically decreased disease activity (based on BASDAI and ASDAS-CRP questionnaires), while calprotectin levels were significantly decreased in both AS and nr-axSpA patients. On the other hand, in the recent study of Husakova et al. [47], disease activity, based on relative questionnaires, statistically improved after 6 months of exercise training only in a group of nr-axSpA patients. In this study, neither was the control group included nor the functional capacity evaluated.

On the contrary, in our study, a long-term Pilates exercise training program combined with walking led to significant improvements in disease activity and functional capacity in patients with nr-axSpA. Our study revealed favorable improvements in BASDAI, ASDAS-CRP, and functional capacity only in the exercised patients. Based on the existing literature on the AS population, exercise may lead to significant immune, metabolic, and endocrine alterations [47]. However, there is a lack of evidence regarding the efficacy of rehabilitation and exercise training in the early stages of axSpA, as only a few studies have examined the effects of exercise in AS by using different physical therapy approaches, and their results are, to some extent, controversial [9]. In the study of Escalas et al. [48], by analyzing data from the DESIR cohort, a 6-month physical therapy (with at least eight sessions of exercise during that period) did not improve functional capacity in patients with early axSpA, concluding that exercise may not be beneficial in the early stages of the disease [9,48]. Dissimilarly, Rosu et al. [16] observed that a 24-week McKenzie or classic kinetic training in 52 patients with early AS statistically improved pain, disease activity, and score in the Bath Ankylosing Spondylitis Functional Index (BASFI) test.

Our study is the first to evaluate the effects of long-term exercise training on cardiorespiratory fitness in patients with nr-axSpA. We found significant improvements in aerobic capacity, as estimated by VO_2_peak and exercise time. According to Sveaas et al. [49], a 12-week high-intensity interval exercise showed promising effects of cardiorespiratory and strength exercises, fatigue, and patients’ ability to engage with daily activities. Niederman et al. [50] showed that a 12-week cardiovascular training program led to a significant fitness level increase, while Nolte et al. [51] found that a long-term multimodal exercise intervention (with land, swimming, and breathing exercises) can improve relative and absolute VO_2_max, BASDAI, and bath ankylosing spondylitis metrology index (BASMI) in the exercise group, compared to the control group. Similarly, in the study of Jennings et al. [52], a 12-week aerobic and stretching exercise program had favorable improvements in aerobic capacity and distance walking, as estimated by a 6 min walking test (6MWT) in patients with AS. Moreover, Taskin et al. [53] noticed that a 12-week aerobic training program in 17 patients with AS was an effective remedy strategy to increase respiratory muscle strength and exercise capacity.

Pilates exercise is an effective type of exercise for healthy adults and patients with AS. Pilates, via breathing techniques, produces satisfactory intra-abdominal pressure to stabilize the lower back. At the same time, different types of Pilates exercises strengthen muscles (especially abdominal, gluteal, and paraspinal muscles), improve spinal support, and help patients with AS to stand upright and relieve lower back pain [54]. Moreover, the intensity of neuromuscular stimulation achieved during Pilates may improve cardiorespiratory fitness by increasing HRmax and VO_2_max for healthy individuals and patients with chronic conditions [55]. From a cellular point of view, recent studies on randomized controlled trials (RCTs) support that long-term Pilates exercise may upregulate mRNA gene expression of GPX1 and SOD2 enzymes, reducing inflammatory chemical markers such as IL-8 and CRP levels and leading to anti-inflammatory benefits [55]. Our study revealed that a 6-month home-based Pilates training program, combined with walking, results in better functional capacity (as measured with TUG, SR, 5×STS, BSL, and BSR tests in Group A) and lower disease activity in patients with nr-axSpA. Based on the physiological [54] and cellular [55] mechanisms of Pilates, we assume these results are mainly due to the Pilates training effect. Nevertheless, combining Pilates exercise with aerobic training can be more effective than aerobic exercise itself. In a recently published study in 2023, Oksüz and Unal [17] showed that combined clinical Pilates and aerobic training resulted in statistically significant improvements in 6MWT, SR, BSL and BSR, BASDAI, and back muscle strength compared to aerobic exercise alone. These results are in agreement with the findings of our study, where our patients were advised to walk the days they did not have a Pilates session. Until recently, no study had investigated the effects of aerobic exercise training alone in patients with AS. To our knowledge, Basakci Calik et al. [56] were the first to find that after a 12-week aerobic exercise training program, patients with AS had statistically improved VO_2_max, BASDAI, and 6MWT, but not BASFI. However, according to the literature, aerobic and strengthening exercise may not improve disease activity, spinal mobility, and even the cardiorespiratory adaptations that may result [52,57]. On the contrary, several studies have shown that Pilates exercise may benefit patients with AS, leading to the desired outcomes, such as pain relief, spinal mobility, pulmonary function improvement, reduced musculoskeletal disorders, and others [44,58]. Altan et al. [15] were the first to show that a 12-week Pilates training program can statistically increase BASDAI, BASFI, quality of life, and chest expansion in patients with AS, indicating that Pilates may be a valuable therapeutic choice in this patient population. Acar et al. [59] showed that an 8-week Pilates exercise training improved functional capacity, balance, disease activity, and quality of life in patients with AS. Similarly, Rodríguez López et al. [60] showed that 1 year of Pilates exercise training statistically increased chest expansion, BASDAI, and patients’ function as measured with BASFI. Furthermore, Bağlan Yentü et al. [14] showed that a long-term Pilates or strengthening exercise program significantly improves respiratory indices such as forced vital capacity (FVC), forced expiratory volume in one second (FEV1), and disease activity in patients with AS. Similarly, Gandomi et al. [61] found that either aqua Pilates or aqua stretch had equal results in BASFI, 40 min walking test (MWT), and pain. According to these results, clinical or home-based Pilates exercise training may be more adaptive for patients with axSpA and nr-axSpA.

Moreover, our study revealed significant correlations between BASDAI and TUG, 5×STS and VO_2_peak, and between ASDAS and 5×STS, SR, and VO_2_peak. The systematic review of Ingram et al. [62] found that functional disability was constantly and negatively associated with physical activity indices in patients with AS. In comparison, negative associations were noticed between physical activity, age, anti-TNFa therapy, and being unmotivated for exercise participation. In contrast, positive correlations were noted among physical activity and VO_2_max, modified Schober test and cervical rotation (tools for assessing spinal mobility), and motivation and quality of life [62]. Moreover, in the study of Arends et al. [63], daily physical activity (assessed with an accelerometer) was significantly correlated with disease activity (BASDAI and ASDAS), BASFI, International Physical Activity Questionnaire (IPAQ), and quality of life.

As mentioned above, physical activity and exercise in patients with AS are crucial in interventions for disease management, as increased physical levels reduce disease activity, pain, and immobility and improve functional capacity and cardiorespiratory fitness [64]. However, it is unknown whether the effects of exercise arise systemically (e.g., anti-inflammatory) or locally (e.g., enthesis). Still, excessive exercise training in patients with AS could possibly and paradoxically lead to entheseal site microdamage and aggravate disease outcomes. The paradox of the benefit and damage of exercise in axSpA has been called the ‘Goldilocks zone’, and it tries to explain the potential role of mechanical stress in AS pathogenesis. Further investigation is required to understand the changes in enthesis sites in patients with nr-axSpA after exercising [7].

To sum up, our study has several strengths. Firstly, this study was the first that evaluated the effects of home-based Pilates exercise training with walking in patients with nr-axSpA, as there is limited evidence of its effectiveness in this population. Secondly, this study revealed statistically significant correlations between disease activity indexes (BASDAI and ASDA) and several functional and aerobic capacity indicators. These results indicate that long-term Pilates exercise training is strongly correlated with lower disease activity levels and better functional and aerobic capacity. Thirdly, this trial was held entirely via an online platform, which is also important, as the patients’ adherence to exercise was high, and it also provided them with individuality and autonomy. The study also has limitations due to a small sample size, which was primarily caused by challenges in recruiting patients for long-term Pilates exercise training. Moreover, we neither compared the effects of home-based Pilates alone versus combined Pilates with aerobic exercise nor did we investigate the possible differences between these types of exercise on disease activity, functional capacity, and cardiorespiratory fitness. Larger randomized clinical trials are greatly needed in patients with nr-axSpA to further investigate the link between the disease and exercise cardiorespiratory and functional outcomes [47].

## 5. Conclusions

In conclusion, long-term Pilates exercise training with walking in patients with nr-axSpA had favorable effects on disease activity, cardiorespiratory fitness, and functional capacity. Pilates exercise is a valuable therapeutic option for this population that requires further investigation.

## Figures and Tables

**Figure 1 jfmk-08-00140-f001:**
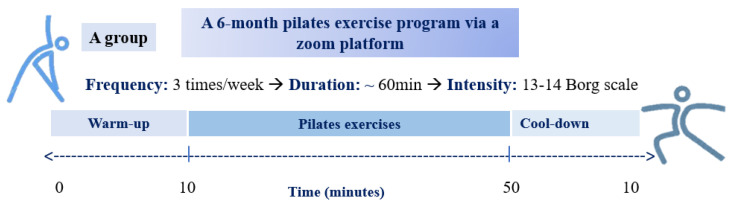
The Pilates exercise program.

**Figure 2 jfmk-08-00140-f002:**
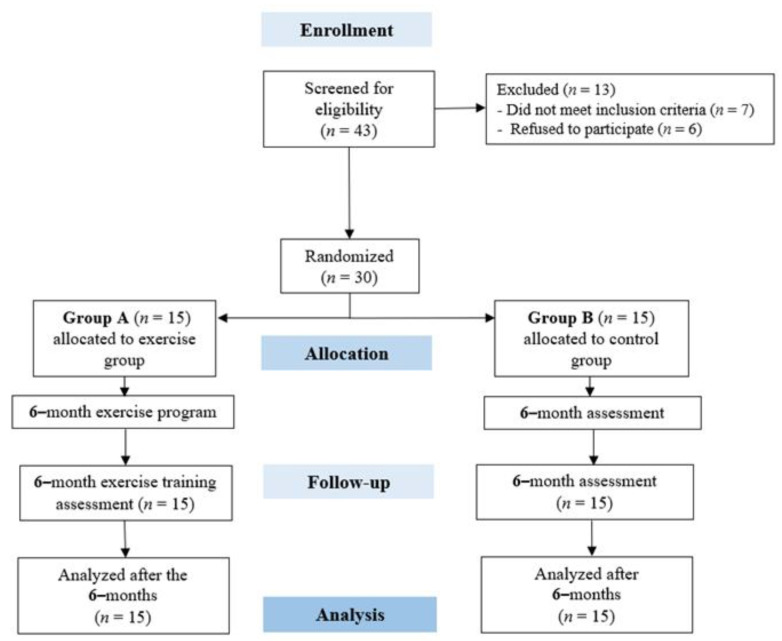
CONSORT diagram of the study design.

**Figure 3 jfmk-08-00140-f003:**
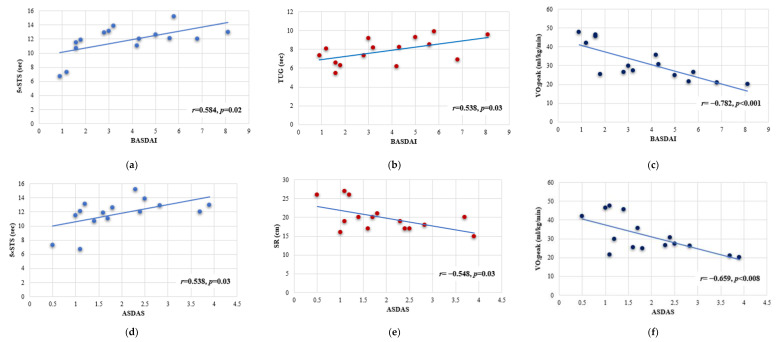
Linear regression analysis between functional capacity and disease activity indicators. The (**a**–**c**) scatter charts represent the correlation between the Bath Anky-losing Spondylitis Disease Activity Index (BASDAI) and timed up and go test (TUG), 5 times sit-to-stand test (5×STS), and maximum oxygen consumption (VO2peak). The (**d**–**f**) scatter charts represent the correlation between the Ankylosing Spondylitis Disease Activity Score (ASDAS) and 5×STS and the correlation between sit-and-reach test (SR) and VO2peak.

**Table 1 jfmk-08-00140-t001:** Clinical characteristics of patients with nr-axSpA.

	Group A (*n*_A_ = 15)			Group B (*n*_B_ = 15)			A vs. B Group	
	Baseline	Follow-Up	*p*-Value	Baseline	Follow-Up	*p*-Value	Pre	Post
Sex (female/male)	14/1	-		13/2	-		*p* = 0.29	
Age (years)	43.73 ± 9.81	-		49.33 ± 10.01	-		*p* = 0.13	
Height (cm)	1.63 ± 0.05	-		1.63 ± 0.06	-		*p* = 0.95	
Weight (kg)	76.72 ± 21.92	76.48 ± 21.05	*p* = 0.81	79.10 ± 16.92	78.03 ± 16.54	*p* = 0.11	*p* = 0.74	*p* = 0.28
BMI (kg/m^2^)	28.54 ± 7.88	28.44 ± 7.50	*p* = 0.80	29.42 ± 6.00	29.00 ± 5.73	*p* = 0.09	*p* = 0.95	*p* = 0.24
CRP (mg/L)	2.55 ± 2.51	2.30 ± 2.15	*p* = 0.73	2.59 ± 1.92	2.44 ± 1.56	*p* = 0.58	*p* = 0.96	*p* = 0.84
ESR (mm/h)	20.46 ± 13.02	18.80 ± 3.36	*p* = 0.60	19.93 ± 7.62	20.26 ± 7.61	*p* = 0.67	*p* = 0.89	*p* = 0.27

Note: nr-axSpA: non-radiologically confirmed axial spondylitis; CRP: C-reactive protein; ESR: Erythrocyte Sedimentation Rate; and BMI: Body Mass Index. Data are expressed as mean ± SD; Significant at the 0.05 level (*p* < 0.05).

**Table 2 jfmk-08-00140-t002:** Cardiorespiratory efficiency at baseline and the end of the study.

	Group A				Group B				A vs. B Group			
	Baseline	Follow-Up	*p*-Value	Intra-Observer Variability ICC (95% CI)	Baseline	Follow-Up	*p*-Value	Intra-Observer Variability ICC (95% CI)	Pre	Inter-Observer Variability ICC (95% CI)	Post	Inter-Observer Variability ICC (95% CI)
HRrest (bpm)	83.66 ± 12.82	80.00 ± 11.62	*p* = 0.02	0.78 (0.34/0.92)	83.80 ± 10.26	84.40 ± 9.61	*p* = 0.73	−0.36 (−0.73/0.15)	*p* = 0.97	0.28 (−1.14/0.75)	*p* = 0.001	0.76 (0.33/0.91)
SBPrest (mmHg)	119.66 ± 14.45	117.13 ± 8.55	*p* = 0.74	0.45 (−0.61/0.81)	120.33 ± 9.53	121.00 ± 9.29	*p* = 0.81	0.10 (−0.41/0.57)	*p* = 0.88	−0.18 (−2.51/0.60)	*p* = 0.15	0.01 (−1.96/0.66)
DBPrest (mmHg)	77.00 ± 8.40	73.33 ± 7.94	*p* = 0.03	0.70 (0.21/0.85)	76.33 ± 7.89	76.00 ± 8.06	*p* = 0.83	0.14 (−0.38/0.59)	*p* = 0.79	0.46 (−0.58/0.82)	*p* = 0.31	0.54 (−0.37/0.84)
Time (min)	7.81 ± 1.85	9.88 ± 2.61	*p* = 0.001	0.76 (0.31/0.92)	7.22 ± 1.19	7.19 ± 1.41	*p* = 0.92	0.17 (−0.34/0.62)	*p* = 0.21	0.54 (−0.35/0.84)	*p* = 0.001	0.72 (0.16/0.90)
METs	7.46 ± 2.88	9.16 ± 2.73	*p* = 0.02	0.74 (0.23/0.91)	7.44 ± 1.28	7.12 ± 1.07	*p* = 0.06	0.22 (−0.30/0.65)	*p* = 0.97	0.25 (−1.23/0.74)	*p* = 0.008	0.56 (0.10/0.82)
VO_2_peak (mL/kg/min)	26.33 ± 9.87	31.46 ± 9.63	*p* = 0.04	0.74 (0.23/0.91)	26.13 ± 4.43	25.08 ± 2.75	*p* = 0.13	0.31 (−0.22/0.69)	*p* = 0.94	0.19 (−1.41/0.72)	*p* = 0.01	0.77 (0.32/0.92)
VO_2_/HRmax	12.68 ± 2.73	13.93 ± 2.76	*p* = 0.04	0.64 (0.47/0.88)	12.66 ± 2.02	12.13 ± 2.06	*p* = 0.06	0.16 (−0.36/0.61)	*p* = 0.99	−1.17 (−5.46/0.27)	*p* = 0.04	0.92 (0.78/0.97)
VE/VO_2_max	29.60 ± 4.11	28.80 ± 6.48	*p* = 0.73	−1.03 (−5.07/0.31)	29.53 ± 4.47	30.26 ± 4.97	*p* = 0.46	0.42 (−0.09/0.76)	*p* = 0.96	0.11 (−1.62/0.70)	*p* = 0.75	0.29 (−1.10/0.76)
VE/VCO_2_max	27.73 ± 3.28	26.33 ± 2.09	*p* = 0.18	−0.15 (−2.44/0.61)	27.33 ± 4.46	27.46 ± 3.62	*p* = 0.92	0.06 (−0.04/0.54)	*p* = 0.71	0.59 (−0.20/0.86)	*p* = 0.35	0.40 (−0.77/0.80)
HRmax (bpm)	158.60 ± 21.65	171.06 ± 19.41	*p* = 0.03	0.68 (0.05/0.89)	157.93 ± 18.26	157.60 ± 19.74	*p* = 0.89	0.37 (−0.14/0.73)	*p* = 0.87	0.30 (−0.22/0.69)	*p* = 0.05	0.36 (−0.15/0.73)
SBPmax (mmHg)	155.33 ± 21.99	147.33 ± 16.02	*p* = 0.04	0.86 (0.60/0.95)	155.66 ± 18.82	155.60 ± 18.11	*p* = 0.97	0.1 (−0.31/0.60)	*p* = 0.96	0.30 (−1.07/0.76)	*p* = 0.18	0.40 (−0.78/0.79)
DBPmax (mmHg)	76.33 ± 9.34	73.33 ± 7.94	*p* = 0.24	0.55 (−0.31/0.85)	76.66 ± 8.99	77.00 ± 8.82	*p* = 0.86	0.30 (−0.23/0.70)	*p* = 0.97	0.36 (−0.87/0.78)	*p* = 0.19	0.25 (−0.28/0.66)

Note: ICC: intraclass correlation coefficient; 95% CI: 95% confidence interval (lower bound/upper bound); METs: metabolic equivalents for physical activity; VO_2_peak: maximum oxygen consumption; VO_2_/HRmax: ratio between VO_2_ and maximum heart rate; VE/VO_2_max: ventilatory equivalents for oxygen; VE/VCO_2_max: ventilatory equivalents for carbon dioxide; HR: heart rate; SBP: systolic blood pressure; and DBP: diastolic blood pressure. Data are expressed as mean ± SD; *p* < 0.05: baseline vs. 6 months follow-up; *p* < 0.05: Group A vs. B.

**Table 3 jfmk-08-00140-t003:** Functional capacity and disease activity assessment at baseline and the end of the study.

	Group A				Group B				A vs. B Group			
	Baseline	Follow-Up	*p*-Value	Intra-Observer Variability ICC (95% CI)	Baseline	Follow-Up	*p*-Value	Intra-Observer Variability ICC (95% CI)	Pre	Inter-Observer Variability ICC (95% CI)	Post	Inter-Observer Variability ICC (95% CI)
TUG (s)	9.35 ± 1.44	7.81 ± 1.34	*p* = 0.001	0.71 (0.15/0.90)	9.17 ± 1.08	9.71 ± 1.40	*p* = 0.14	0.19 (−1.40/0.72)	*p* = 0.69	0.27 (−1.47/0.71)	*p* = 0.001	0.57 (0.11/0.83)
SR (cm)	17.20 ± 3.93	19.86 ± 3.73	*p* = 0.001	0.98 (0.95/0.99)	17.20 ± 5.22	16.73 ± 4.65	*p* = 0.25	0.10 (−0.41/0.57)	*p* = 0.99	0.59 (−0.20/0.86)	*p* = 0.007	0.73 (0.20/0.91)
BSR (cm)	−1.93 ± 5.61	−0.73 ± 3.78	*p* = 0.03	0.89 (0.68/0.96)	−2.00 ± 5.96	−2.13 ± 5.71	*p* = 0.86	−0.05 (−0.53/0.45)	*p* = 0.97	−0.25 (−2.73/0.57)	*p* = 0.49	−0.70 (−4.06/0.42)
BSL (cm)	−3.20 ± 7.10	−2.26 ± 6.09	*p* = 0.04	0.98 (0.94/0.99)	−3.46 ± 10.64	−3.40 ± 9.47	*p* = 0.88	−0.26 (−0.67/0.27)	*p* = 0.06	−0.25 (−2.75/0.57)	*p* = 0.71	−0.10 (−2.30/0.62)
5×STS (s)	13.99 ± 2.59	11.73 ± 2.21	*p* < 0.001	0.88 (0.64/0.96)	13.43 ± 3.16	13.35 ± 4.08	*p* = 0.89	0.13 (−0.48/0.50)	*p* = 0.58	0.02 (−1.90/0.67)	*p* = 0.001	0.86 (0.61/0.95)
BASDAI	4.87 ± 2.32	3.72 ± 2.19	*p* = 0.04	0.62 (0.43/0.86)	4.29 ± 2.31	4.65 ± 2.34	*p* = 0.10	0.30 (−1.06/0.76)	*p* = 0.51	−0.06 (−2.17/0.64)	*p* = 0.04	0.73 (0.20/0.84)
ASDAS	2.66 ± 0.97	1.93 ± 0.99	*p* = 0.04	0.97 (0.91/0.99)	2.45 ± 1.15	2.52 ± 1.23	*p* = 0.19	0.18 (−0.34/0.62)	*p* = 0.63	−0.47 (−3.39/0.50)	*p* = 0.03	0.88 (0.67/0.91)

Note: TUG: timed up and go test; SR: sit-and-reach test; BSR: back scratch test for the right arm; BSL: back scratch test for the left arm; 5×STS: 5 times sit-to-stand test; BASDAI: Bath Ankylosing Spondylitis Disease Activity Index; ASDAS: Ankylosing Spondylitis Disease Activity Score; ICC: intraclass correlation coefficient; and 95% CI: 95% confidence interval (lower bound/upper bound). Data are expressed as mean ± SD; *p* < 0.05: baseline vs. 6 months follow-up; *p* < 0.05: Group A vs.B.

## Data Availability

In the case where access to this study’s data is needed, please contact the corresponding author. The data are not currently available to the public as they are subject to ethical restrictions.

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
