# Peer review of "The Effects of Pilates Exercise Training Combined with Walking on Cardiorespiratory Fitness, Functional Capacity, and Disease Activity in Patients with Non-Radiologically Confirmed Axial Spondylitis"

_jfmk, 2023, doi:10.3390/jfmk8040140_

Round 1
Reviewer 1 Report
This is a very interesting work, made in a very simple way. Methodologically ok.
However i believe that could (must) be improved:
1) A conclusion is a strong and obvious result from a study. In the abstract i sugest that authors state (last phrase) : "to sum up.... improved after a long-term...." instead of "to sum up...can be improved after a long-ter,,,,". I believe that results were good and patients have improved! So, it seems that this typi of exercise IN FACT has improved physical capacity (in many components) of these patients. Just be more powerfulness in your conclusions.
2)The inclusion criteria is some way vague. Only patients with the genetic diagnostic of nr-axSpA, with pain? Why should they do Pilates or other exercise? Have this population other symptoms? No other signs or symptoms? Have they vertebral disconfort or pain due to other reasons than nr-axSpA? Please state that.
3) To achieve the sample size we should know the population. The sample size of 30 participants was based on wat population? Al SpA of the country? Of the city? What was?
4) In fig 1 is not clear the duration of Pilates component. In the text authors stated that it should be a duration of 40min. However in the figure presented, in the way that authors present it seems that this part of the session is between 10min and 40 min (so, is 30 min). I believe that authors could improve this figure.
5) I do not find the value for the adeshion of patients. What was the percentage of participation? How many exercise sessions have they done? All? 80%? 50%? This is crucial for the results!
6) I believe that the most important to be improved is indeed the home-based program (sessions by zoom) methodology. Why have pilates sessions by zoom? Any special reason? How have authors supervised in the zoom, the correct way for exercises? How was done the feed-back's?
7) Authors do not discuss if improvements are a consequence of Pilates, or Pilates+walking or...if is only a consequence of walking everydays? This is very important for a good and honest discussion and in my oppinion crucial for this manuscript
Minor correction
8) Try to improve the references. Only 9/46 are from 2021-2023.
Reviewer 2 Report
The manuscript has been satisfactorily presented, standing out for its innovative and relevant content. However, there are some aspects that can be improved in order to further enhance the quality of the material. Please find below some suggestions/comments:
Introduction
1) Although the present study aimed to assess the effects of Pilates exercise, the authors poorly explored in the introduction section the potential effects of this training modality on the outcomes of interest.
2) Authors stated that studies on nr-axSpA are limited, however what it is known on effects of training (i.e., walking or Pilates) in axSpA. Besides, is the only difference between the conditions the damage visibility on X-Ray? What about the differences in terms of symptoms?
Methods
3) Authors stated that patients who did not have chronic comorbidities would be included in the study and later it is described that patients with chronic comorbidities were excluded. There is an overlap of information.
4) Sample size section: Authors “hypothesized significance differences between groups A and B”. However, the meaning of groups A and B has not been described before.
5) Cardiopulmonary test: Line 123 – Did you mean “Oxygen pulse (VO2/HRmax)”? Moreover, please include reference for the end-point during maximal exercise (i.e., RER and VO2 plateau).
6) Time up and go test: The cutting point referenced was determined for older adults, however, authors included patients aged <65 years old.
7) 5xSTS test: Is there any reference values for this test interpretation?
8) Authors described the Pilates exercises, however it is not clear if the protocol was the same for all participants.
9) About the exercise protocol, the patients were instructed to walk for about 30min, was this activity controlled/monitored? Instructions on the intensity was provided?
Results and Discussion
10) Figure 3: Consider including R and P values in each graphic.
11) Discussion: Authors should explore more deeply the possible physiological mechanisms involved in the Pilates practice and increase on cardiorespiratory fitness, functional activity, and reduction of disease activity, as well as how they are related. Not only state that previous studies found increase/decrease on those variables but what are the possible mechanisms.
12) How did authors ensure that benefits reached with the protocol was due to Pilates or walking?
Conclusion
13) Authors should include that not only Pilates, but Pilates associated with walking had impact in the patients with nr-axSpA.
Reviewer 3 Report
This manuscript describes a study where individuals with non-radiologically confirmed axial spondylitis were randomized to a control group and a pilates home-exercise group for 6-months with a pre- post- study design. While the results are interesting, there are a some things that the authors need to consider to strengthen their manuscript.
1. In the introduction, please include a hypothesis statement for the study.
2. In the materials and methods section, please include a list of all exclusion criteria for the study instead of just stating 'etc.'.
3. How was nr-axSpA diagnosed in the patients that were part of this study?
4. It is stated that the patients did not use medication during the study but in their medical histories it was stated that painkillers/anti-inflammatory drugs were recorded for use. Did the study require patients to stay off of prescription or non-prescription drugs for the duration of the study? What happened if the patient had a flare-up during the study?
5. Under 2.3, it is stated that participants were blinded to group allocation. How were participants blinded to an exercise intervention when they are actually participating in it?
6. Section 2.4 in the methods needs to be drastically expanded on to improve the repeatability of the study.
7. Section 2.5 states that one of the criteria for reaching an endpoint with the cardiorespiratory exercise testing was an oxygen plateau. How was this oxygen plateau defined?
8. I believe the correct terminology for the TUG is 'timed up and go' test not 'time up and go'.
9. Section 2.10 - BASDAI cannot be defined as >/= 4 and </= 4 - this is not correct. I believe the correct method would be >/= 4 is high disease activity and < 4 is low disease activity.
10. Section 2.12 - please provide more specific information on the exercises and exercise progressions that were used in the study throughout the 6-months to ensure repeatability of the study if ever done again.
Minor comments:
11. L 332 - do you mean mobility?
12. L 346-347 - just make a statement instead of asking a question in the discussion.
13. L 366 - systemically not systematically.
14. L 373-374 - remove overall word from the sentence to improve clarity.
Minor spell check required but otherwise, well-written.
Round 2
Reviewer 1 Report
Thank you to the authors for the good revision that they made to the manuscript.
I fully accept this manuscript for publication with the corrections that they have made.
Reviewer 2 Report
The authors have made substantial changes to the manuscript to address most of my concerns. Nothing more to add.
Reviewer 3 Report
Thank you for addressing my queries and for the detailed response of what was included in the exercise routine for the participants in this group.